# The Therapeutic Potential of Kaemferol and Other Naturally Occurring Polyphenols Might Be Modulated by Nrf2-ARE Signaling Pathway: Current Status and Future Direction

**DOI:** 10.3390/molecules27134145

**Published:** 2022-06-28

**Authors:** Yaseen Hussain, Haroon Khan, Khalaf F. Alsharif, Amjad Hayat Khan, Michael Aschner, Luciano Saso

**Affiliations:** 1College of Pharmaceutical Sciences, Soochow University, Suzhou 215123, China; pharmycc@gmail.com; 2Department of Pharmacy, Bashir Institute of Health Sciences, Islamabad 45400, Pakistan; 3Department of Pharmacy, Abdul Wali Khan University, Mardan 23200, Pakistan; 4Department of Clinical Laboratory, College of Applied Medical Science, Taif University, P.O. Box 11099, Taif 21944, Saudi Arabia; alsharif@tu.edu.sa; 5Department of Allied Health Sciences, Bashir Institute of Health Sciences, Islamabad 45400, Pakistan; ahkhan086@gmail.com; 6Department of Molecular Pharmacology, Albert Einstein College of Medicine, New York, NY 10463, USA; michael.aschner@einsteinmed.org; 7Department of Physiology and Pharmacology “Vittorio Erspamer”, Sapienza University, P.le Aldo Moro 5, 00185 Rome, Italy

**Keywords:** kaempferol, Nrf2 modulation, targeting Nrf2 modulation, therapeutic outcomes

## Abstract

Kaempferol is a natural flavonoid, which has been widely investigated in the treatment of cancer, cardiovascular diseases, metabolic complications, and neurological disorders. Nrf2 (nuclear factor erythroid 2-related factor 2) is a transcription factor involved in mediating carcinogenesis and other ailments, playing an important role in regulating oxidative stress. The activation of Nrf2 results in the expression of proteins and cytoprotective enzymes, which provide cellular protection against reactive oxygen species. Phytochemicals, either alone or in combination, have been used to modulate Nrf2 in cancer and other ailments. Among them, kaempferol has been recently explored for its anti-cancer and other anti-disease therapeutic efficacy, targeting Nrf2 modulation. In combating cancer, diabetic complications, metabolic disorders, and neurological disorders, kaempferol has been shown to regulate Nrf2 and reduce redox homeostasis. In this context, this review article highlights the current status of the therapeutic potential of kaempferol by targeting Nrf2 modulation in cancer, diabetic complications, neurological disorders, and cardiovascular disorders. In addition, we provide future perspectives on kaempferol targeting Nrf2 modulation as a potential therapeutic approach.

## 1. Introduction

Nrf2 (nuclear factor erythroid 2-related factor 2) is a transcription factor involved in carcinogenesis and other diseases, playing an important role in modulating redox status and inflammation [1]. In the context of oncology, Nrf2 targeting has been considered an important chemotherapy and chemoprevention strategy [2]. It has been demonstrated that the modulation of Nrf2 leads to cell protection from cancer promotion and initiation [3]. The enhanced activation and overexpression of Nrf2 renders the cells resistant to radiotherapy and conventional chemotherapy [4]. In addition, the quenching and production of reactive oxygen species and free radicals-related intracellular/extracellular signaling needs to be strictly regulated upon cellular damage [5]. Nrf2 also plays a critical role in cardiovascular disease [6], and Nrf2 targeted activation might open new avenues in cardiovascular disease therapeutics. In addition to carcinogenesis and cardiovascular disease, studies have shown the role Nrf2 in diabetes complications and neurological disorders [7,8,9].

Natural products based phytochemicals increase cancer cell death primarily by regulating Nrf2 [10], leading to pro-autophagy, oxidative stress, pro-apoptosis and inhibition of expression of cytoprotective genes [11]. Several phytochemicals exert their anticancer or chemo preventive role by Nrf2 modulation [12]. These include quercetin, curcumin, and resveratrol [13]. Natural Nrf2 inhibitors, such as luteolin, malabaricone, wogonin, and ascorbic acid reduce the production of Nrf2 and thus offer chemo sensitizer and anti-carcinogenic activity in tumors [14].

Kaempferol is a natural flavonoid extracted from tea. Other sources include kale, grapes, citrus fruits, vegetables, beans, apple, broccoli, and tomatoes. Physically, it is crystalline in nature with pure yellow color. It is slightly soluble in water [15], and it has shown efficacy as an anticancer, anti-inflammatory, and anti-arthritic compound, given its antioxidant and antiviral activity [16,17,18].

This review article highlights the therapeutic role of kaempferol in cancer, cardiovascular diseases, diabetic complications, and neurological disorders by targeting Nrf2 modulation. The current status of kaempferol targeting Nrf2 modulation is underpinned along with the future directions that need further exploration of this versatile phytochemical.

## 2. Nrf2 Activation/Inhibition

Nrf2 (nuclear factor erythroid 2-related factor 2) activation results in the expression of several proteins and cytoprotective enzymes that provide cellular protection against reactive oxygen species (ROS), thus offering cytoprotection [19]. Nrf2 is thus believed to be the first defense line against agents that cause cancer initiation and promotion.

Nrf2 under normal conditions is sequestered in cytoplasm and attached to its repressor receptor known as Keap1 (Kelch-like ECH-associated protein 1) [20]. Beyond its function in cell protection, it is implicated in a wide network regulating anti-inflammatory response and metabolic reprogramming, i.e., a key regulator of cell fate and a strategic player in the control of cell transformation and response to viral infections [21]. Several phytochemicals, such as isothiocyanate, act as Nrf2 activators, binding to the SH group of keap-1 protein and, in turn, leading to inhibition of Nrf2 degradation. Under oxidant stress conditions, Nrf2 is translocated to the nucleus, where it induces a variety of genes involved in the antioxidant defense [22].

Several protein kinases, i.e., AMP-activated protein kinases, tyrosine protein kinases, and mitogen-activated protein kinases, lead to Nrf2 post-translational modifications that trigger the release of Nrf2. These protein kinases phosphorylate Nrf2, which, in turn, modulates its activity and stability. Such phosphorylation and consecutive activation of Nrf2 leads to therapeutic potential in inflammatory disorders [23]. The p62 protein is an autophagy receptor for the degradation of proteins and mitochondria. Keap 1 is degraded after interaction with p62 protein via autophagy, in turn leading to stabilization of Nrf2 [24,25]. On the other hand, increased phosphorylation of p62 and autophagy impairment lead to Nrf2 activation, promoting proliferation of cancer cells. Nrf2 protein levels are also regulated by epigenetic mechanisms, such as microRNA and methylation of Keap 1 promoters [26].

Nrf2 also contributes to chemoresistance, proliferation, and invasion [27]. Thus, Nrf2 pro-oncogenic activation in cancer is associated with mechanisms that involve both genetic and epigenetic alterations [28]. Nrf2 leads to cancer cell proliferation following certain metabolic re-programming [29]. In normal cells, Nrf2 signaling cascades afford defensive mechanisms, while attenuated Nrf2 levels lead to tumorigenesis [30,31]. For example, Nrf2-null mice were found more susceptible to exposure to carcinogens [32]. Nrf2 is the key transcription factor regulating antioxidant and xenobiotic exposure response. When oxidative stress increases, Nrf2 translocates to the cell nucleus and forms heterodimer with small Maf (sMaf) proteins. Nrf2/sMaf heterodimer binds specifically to a cis-acting enhancer called antioxidant response element (ARE) and initiates transcription of genes encoding antioxidant and detoxification proteins [33]. Nrf2 modulation in cancer cells is schematically illustrated in Figure 1.

In human lung cancer, persistent Nrf2 modulation has been shown to result in toxicity attenuation. Conversely, cellular response to drug therapy and ionizing radiations was enhanced upon Nrf2 knockdown [34]. In tumor metastasis, Nrf2 modulation has been shown to result in inhibition of pro-metastatic transcription factor Bach1 degradation, which resulted in promoting lung cancer [35]. Furthermore, Nrf2 modulation was found to be linked to RhoA gene activation, leading to metastasis and proliferation of breast cancer cells [36]. A reduction in antioxidant defense or excessive production of free radicals has been shown to lead to redox imbalance, in turn inducing cardiovascular diseases.

In diabetes mellitus, Nrf2, in addition to its protective response, exerts other significant functions that contribute to the management of diabetes mellitus [37].

Therefore, Nrf2 targeting may pave new ways in the therapy of life-threatening disorders including diabetes.

## 3. Phytochemical as Nrf2 Modulators

Among the phytochemicals, polyphenols are the most extensively studied and diversified group of phytochemicals [38]. Flavonoids and phenolic acids are classified as two major classes of polyphenols. Hydroxycinammic and hydroxybenzoic acid are classified as main classes of phenolic acid. Further, flavonoids are classified as flavonols, flavones, flavanol, flavonones, and isoflavone [39,40]. Various molecules and signaling pathways implicated in cell death, proliferation, and differentiation are targeted by polyphenols [41,42,43]. The relationship between phytochemicals and Nrf2 signaling is shown in Figure 2. In the prophylaxis of cancer, the cytoprotective role of Nrf2 was elaborated based on expression of various genes through activation of Nrf2 signaling. Nrf2 induction by phytochemicals has been investigated recently, showing Nrf2 activation [44].

Tea components have shown efficacy in in vitro and in vivo disease models, including cancer [45,46,47]. The derivatives of betalamic acid, xanthohumol, betalains, and tannins have been explored to a lesser degree; nonetheless, they do exhibit efficacy in modulating Nrf2 [48]. Tannic acid has been shown in vitro to increase the expression of phase II enzymes downstream of Nrf2 [49]. Betanin was explored in hepatoma-derived HepG2 cell lines in hepatic cancer, showing anticarcinogenic and hepatoprotective effects by Nrf2 activation [50], consistent with findings with xanthohumol. In addition, xanthohumol via activation of Nrf2 signaling has shown anti-inflammatory efficacy in microglial BV2 cells [51]. Induction of the AMPK/GSK3β-Nrf2 pathway was shown to afford protection in acute lung injury [52]. Urosolic acid was investigated in epidermal JB6P+ cells of a mouse skin cancer model, showing Nrf2 activation and increased antioxidant efficacy [53].

Other phytochemicals, such as apigenin and luteolin, were evaluated in NSCLC/A549 cells of non-small cell lung cancer and Bel-740ADM cells of hepatocellular carcinoma. Both reduced protein and Nrf2-mRNA levels, resulting in anti-cancer activity [54,55]. Nrf2 inhibitors were also investigated for cancer treatment. Quassinoid brusatol, a potent inhibitor of Nrf2 that acts via stimulation of Nrf2 poly-ubiquitination, led to reduction in Nrf2 protein levels [28]. Studies in human xenograft myeloid leukemia models showed that quercetin induced apoptosis via reduction in nuclear translocation of Nrf2 and induction of proteasomal degradation via Nrf2 [56]. Details on selected phytochemicals modulating Nrf2 are shown in Table 1.

Combination delivery of phytochemicals has also shown significant effect through modulation of Nrf2. Resveratrol, xanthohumol, and phenethyl isothiocyanate when co-delivered in PANC-1 cells of pancreatic cancer have shown increased DNA and Nrf2 binding along with increased Nrf2 expression [69].

## 4. Pharmacological Actions of Kaempferol

Kaempferol belongs to a class of natural flavonol. Kaempferol (Figure 3) has been shown to reduce the risk of chronic diseases, such as cancer, diabetes, obesity, and liver injury [70,71]. Kaempferol has also been used for its anti-inflammatory potential in various chronic and acute inflammatory conditions, such as acute lung injury, disc degeneration, and colitis [15]. In acute lung injury, kaempferol, has shown efficacy in mitigating pulmonary inflammatory responses and suppress MAPKs and NF-kB signaling pathway [72]. In disc degeneration, kaempferol has been shown to enhance the viability of bone marrow-derived mesenchymal stem cells through increased cell proliferation and LPS (lipopolysaccharides)-induced cell apoptosis [73]. In colitis, kaempferol has been shown to suppress inflammatory activity [74].

In the hippocampus, kaempferols attenuate hippocampus apoptosis and memory defects induced by Cd-Cl2, targeting Akt/mTOR signaling pathway [75]. It also elevates the level of butyrate receptors, tight junction proteins, and transporters in intestinal mucosa, leading to prophylactic treatment of liver injury induced by alcohol [76]. Kaempferol has been also used for prostate cancer treatment. It promotes apoptosis and inhibits cell proliferation [77]. By regulating vasohibin-1 and ERK signaling, kaempferol prevents high glucose-induced injury in retinal ganglion cells [78]. In addition, kaempferol has a bone-sparing effect via mTOR/PI3K/Akt signaling pathway [79,80]. Co-delivery of kaempferol, quercetin, and pterostilbene has been shown to activate Nrf2 signaling pathways synergistically in hepatic cells, resulted and attenuate ROS generation. Such combination leads to increased binding of Nrf2 to ARE [81].

Promoting mitochondrial function by reduction of oxidative stress is a suitable approach for coping with oxidative stress. D-ribose accumulation in mesangial cells leads to the production of ROS and induces advanced glycation end-products, which, in turn, leads to apoptosis. Kaempferol via autophagy repair mechanism has shown to reverse such effects [82]. In acute lung injury, kaempferol has been shown to reduce damage by altering the ubiquitination of TNF receptor-associated factor-6 (TRAF6). In addition, kaempferol via miR-181a upregulation and MAPK/ERK inactivation leads to proliferation suppression of human gastric cancer cells [83].

The neuroprotective effect of kaempferol was evaluated in striatal injury models. Kaempferol provided a neuroprotective effect by maintaining the integrity of the blood-brain barrier, abating neuroinflammation, and downregulation of the toll-like receptor 4 (TLR4) signaling pathway [84]. Such effects were attributed to protein kinase B/β-catenin cascade (AKT) upregulation and was explored in a mouse model [85], showing reduction in DNA fragmentation and increased cell proliferation targeting PI3K/AKT pathway [86]. Kaempferol has also been shown to inhibit uterine fibroid cells [87]. The inflammation and oxidative stress induced by AGE-RAGE/MAPK was significantly reduced by kaempferol in diabetic rats [88]. Finally, kaempferol inhibits the NF-κB pathway activation in osteoarthritis chondrocytes decreasing the level of interleukin-1β-stimulated pro-inflammatory mediators [89].

## 5. Kaemperol Affects Nrf2 Activation in Different Pathological Conditions/Diseases

### 5.1. Cancer

In order to reduce oxidative stress in cancers, many potential signaling and molecular pathways are involved, and kaempferol has been considered an effective agent targeting Nrf2 signaling pathway [90]. In combating cancer, kaempferol regulates transcriptional pathways and restores redox homeostasis, as shown in Figure 4.

The regulation of the Nrf2 signaling pathway affects various factors and other signaling, which, in turn, plays a significant role in cancer [91]. Nrf2 is involved in drug resistance during lung cancer therapy and cancer cell survival [92]. Kaempferol was evaluated for its anti-Nrf2 inhibition potential. Lung cancer A549 and NCIH460 cell lines were used, followed by Nrf2 reporter assay, showing reduction in protein and Nrf2 mRNA levels along with down regulation of Nrf2 target gene transcription. Kaempferol resulted in no change in NFκBp65 and phospho NFκBp65 levels. In addition, kaempferol-induced Nrf2 inhibition resulted in accumulation of reactive oxygen species, sensitizing non-small cell lung cancer cells to apoptosis [93]. Kaempferol-loaded nanoparticles were fabricated and evaluated in a hepatocellular carcinoma model, offering a suitable delivery system for kaempferol during hepatic cancer and Nrf2 signal modulation [94].

Modified Xiaoyao powder—Chinese traditional medicine has been used in the treatment and prevention of breast cancer. Nrf2 was investigated in MCF7 breast cancer cell line followed by determination of chemo preventive action Nrf2 knockdown and Nrf2 wild-type MCF-10A cells, showing upregulation of Nrf2 expression with reduction in oxidative stress. The active compounds were kaempferol and quercetin [95].

### 5.2. Cardiovascular Diseases

Inflammatory responses and oxidative stress are mediators of vascular pathology [96,97]. Kaempferol has been evaluated for its protective action in vascular endothelium in a mouse model. Kaempferol administration in cardiovascular injury significantly increased the level of Nrf2 along with attenuation of antioxidant levels [98]. The proposed mechanism of kaempferol on atherosclerosis is illustrated in Figure 5.

Kaempferol led to reduction in heart Nrf2 level with elevation in the level of Keap-1 mRNA [99]. Several drugs and targets were investigated for the management and treatment of cardiac hypertrophy [100,101,102]. In this context, kaempferol’s effect on collagen accumulation induced by angiotensin II was investigated in C57BL/6 mice, showing that cardiac dysfunction and fibrosis induced by angiotensin II was significantly reduced by kaempferol administration. In addition, the angiotensin II-related oxidative stress and inflammation was prevented by kaempferol through regulation of AMPK/NRF2 signaling pathway [103]. In addition, kaempferol normalized lipid level and blood vessel morphology with suppression of apoptosis and inflammation, secondary to activation of Nrf2 and estrogen receptor coupled with G-proteins [104].

### 5.3. Diabetes Complications

Nrf2 activators are effective in mitigating diabetic complications. Nrf2 play a significant role in insulin sensitivity and beta cell function [105]. Kaempferol has been shown to be effective against diabetic nephropathy secondary to its antioxidant potential [106]. Nrf2 levels have been shown to increase in response to kaempferol and to improve heart function [107]. ARE driven genes transcription plays an important role in the regulation of oxidative stress induced by transient hyperglycemia in normal beta cells [105]. Kaempferol modulates of Nrf2 diabetic complications and delays its progression [108].

### 5.4. Neurological Disorders

Kaempferol has been evaluated for its therapeutic potential in neuroinflammation. Results of the study revealed that Nrf2 binding in microglia to DNA was enhanced by kaempferol, inhibiting neuroinflammation [109]. The neuroprotective effect of kaempferol via Nrf2 regulation was evaluated in a rat model upon chlorpyrifos treatment, a food contaminant and an agricultural pesticide. Results showed a significant protection through kaempferol against neuronal damage, mediated by *GSK3β* and inhibition of Nrf2 induction [110]. Kaempferol also significantly (*p* ≤ 0.001) increased the total antioxidant capacity and improved the memory via elevation of Nrf2 [111].

## 6. Kaempferol Modification during Digestion and Colonic Fermentation

Kaempferol is a poorly absorbed flavonoid. It is also modified during digestion and colonic fermentation. In a recent study, kaempferol-3-glucoside and rutinoside were identified by HPLC. The contents posed modifications at gastric level as compared to intestinal and oral digestion. The rutinosides formed during transport from mouth to intestine showed half reduction in its value, but was stable in the gastrointestinal tract [112]. Under oral and gastrointestinal conditions, the kaempferol glycosides were also found unstable. The conversion of kaempferol glycosides into aglycone kaempferol was also reported following hydrolysis by glycosidases during oral and intestinal digestion. The resultant contents then bind to starch in the digestive tract [113]. Furthermore, another study showed kaempferol glycosides degradation and absorption during gastrointestinal digestion [114]. The Nrf2 interactome is functionally linked to cytoprotection in low-grade stress, chronic inflammation, metabolic alterations, mechanical stress, and ROS formation [115]. Analysis of these molecular profiles suggests alterations of Nrf2 expression and activity as a common mechanism in a subnetwork of diseases referred to as the Nrf2 diseasome. In addition to the role of Nrf2 in various diseases, acute and regular exercise can also induce a state of “low stress” through the production of ROS and stimulation of other mechanoreceptors that activates Nrf2 to modulate endogenous antioxidant systems.

## 7. Conclusions and Future Directions

Kaempferol is a natural flavonoid that offers therapeutic potential. The main sources of kaempferol include tea, kale, spinach, grapes, and gingko biloba leaves. It exhibits the potential of reducing the risk of certain chronic diseases such as cancer, diabetes, obesity, and liver injury. In addition, it has been used in many acute and chronic inflammatory conditions.

The clinical applications of kaempferol are hindered by its low bioavailability. Thus, to improve the clinical efficacy, there is need for structural modifications and novel formulation development. Specific site modification via glycosylation or methylation might be effective because these compounds endogenously occur in plants. Furthermore, the low bioavailability of kaempferol has limited its use in cancer therapy, therefore combination delivery of kaempferol with other anti-cancer drugs. The delivery of kaempferol as nanotechnology scaffolds also offers a versatile choice for cancer and other pathologic treatments.

In modulating Nrf2, several phytochemicals have shown synergistic effects after co-delivery. Kaempferol in the same may be used with other phytochemicals, resembling the combination of phytochemicals in food matrix.

## Figures and Tables

**Figure 1 molecules-27-04145-f001:**
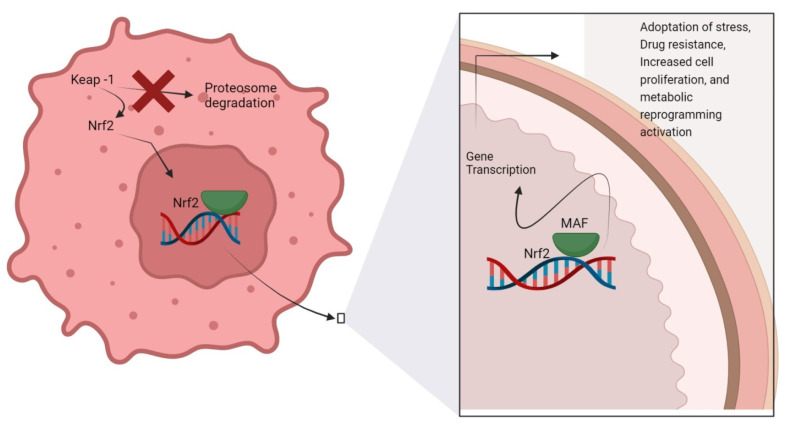
Nrf2 activation/inhibition in cancer cells. Various molecular mechanisms in cancer cells constitute activation of Nrf2, leading to gene expression associated with progression of tumors. The resultant effect is in the form of activation of metabolic reprogramming, enhanced cell proliferation, drug resistance, and adaptation to stress. Nrf2 antioxidant activation also induces an imbalance in the carbon metabolism in cancer. Keap-1; kelch-like ECH-associated proteins, Nrf2; nuclear factor erythroid 2-related factor 2, MAF; musculoaponeurotic fibrosarcoma protein.

**Figure 2 molecules-27-04145-f002:**
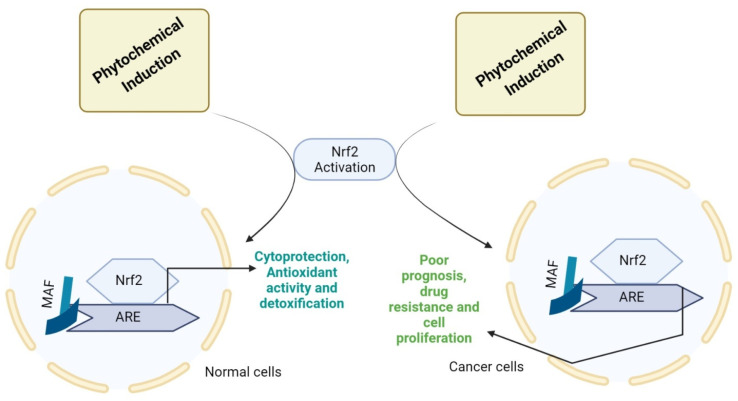
Nrf2 activation by phytochemicals leading to various events. MAF (musculoaponeurotic fibrosarcoma protein) activates antioxidant response genes by interacting with Nrf2 through ARE signaling, indicating that in response to electrophilic and oxidative stresses. In addition, in cancer cells, the imbalance in Nrf2/ARE signaling leads to drug resistance, poor prognosis, and cell proliferation. So, the consecutive Nrf2 activation might cope with the consequences in a positive manner.

**Figure 3 molecules-27-04145-f003:**
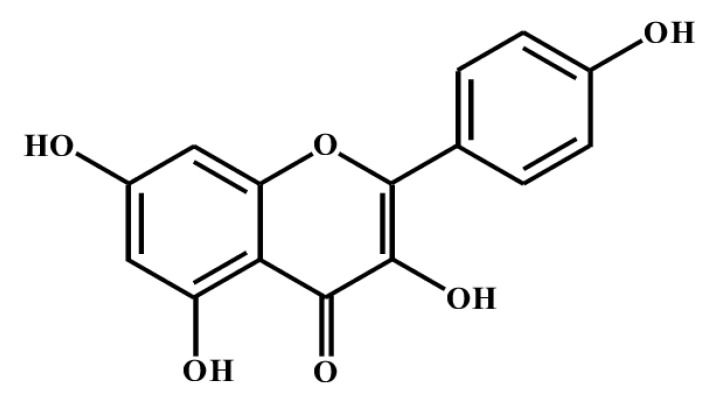
Chemical structure of kaempferol.

**Figure 4 molecules-27-04145-f004:**
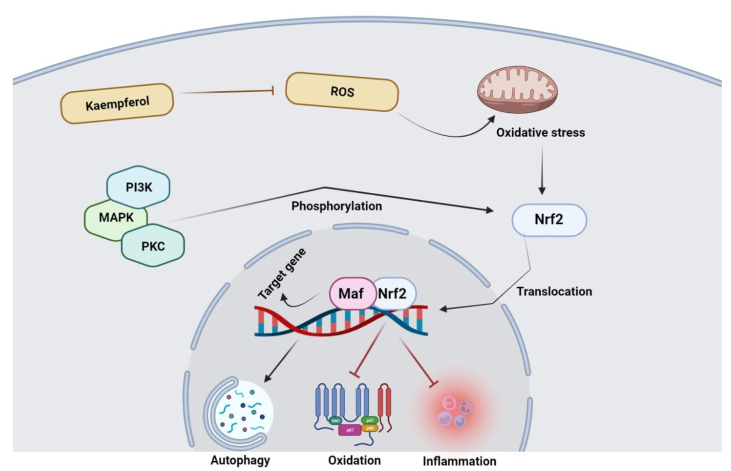
Kaempferol antioxidant potential via modulation of Nrf2. Reactive oxygen species metabolism is inhibited by kaempferol via acting on Nrf2 complex. After disintegration of the Nrf2 complex, it is translocated to the nucleus where it binds to Maf; as a result, the expression of target genes takes place, eventually causing the inhibition of inflammation, oxidation, and induction of autophagy.

**Figure 5 molecules-27-04145-f005:**
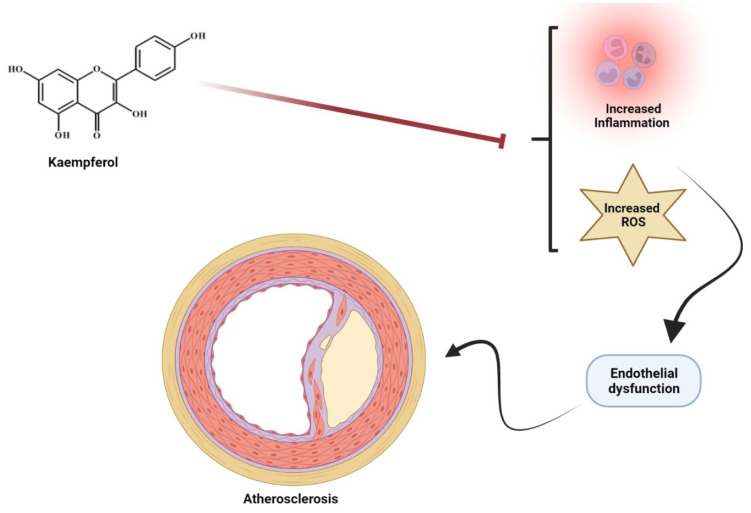
Kaempferol proposed mechanism on atherosclerosis. Kaempferol via inhibition of increased inflammation and reactive oxygen species through modulation of endothelial dysfunction that results in inhibition of atherosclerosis.

**Table 1 molecules-27-04145-t001:** Phytochemicals (non-nutrient) modulating Nrf2.

Phytochemicals	Model	Concentration	Effects on Nrf2	References
Apigenin	HepG2 cells	6.2 µM	Decreased proteins and mRNA levels of Nrf2	[57]
Sappanone	RAW264.7 cells	30 µM	Nrf2 increased nuclear translocation	[58]
Xanthohumol	PANC-1 cells	5–10 µM	Nrf2 and DNA binding along with Nrf2 increased expression	[59]
Sulforaphane	MCF-7 cells	5, 10 and 20 µM	Increased Nrf2 expression	[60]
Oridinin	MG-63 cells, c nude mice	30 mg/kg	Nrf2 decreased nuclear translocation	[61]
Wogonin	MCF-7 cells	60 µM	Decreased Nrf2 expression	[62]
Genistein	Laying Hen model	52.48 mg/hen	Increased Nrf2 expression	[63]
Resveratrol	HaCaT cells	60 µM	Increased nuclear level of Nrf2	[64]
Quercetin	ICR mice (male)	40–80 mg/kg	Mimicked nuclear translocation of Nrf2	[65]
Curcumin	MCF-7 cells	20–40 µM	Nrf2 increased expression	[66]
Piperine	Wistar rats (male)	30–60 mg/kg	Increased Nrf2 expression	[67]
Arctigenin	SD rats (male)	20 mg/kg	Increased SOD expression	[68]

## Data Availability

Not applicable.

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
