# Peer review of "The Therapeutic Potential of Kaemferol and Other Naturally Occurring Polyphenols Might Be Modulated by Nrf2-ARE Signaling Pathway: Current Status and Future Direction"

_molecules, 2022, doi:10.3390/molecules27134145_

Round 1
Reviewer 1 Report
This review authorized by 7 authors from different countries aimed to describe the role of Nrf2 pathway in chemopreventive or therapeutic potential of kaempferol.
However, this manuscript is actually more concentrated on the description of Nrf2 pathway, although not in details. In this regard the mechanisms of its activation, particularly non-canonical, are not well presented or explained. In Figure 1 MAF protein is shown, but its role in Nrf2 is not provided and ARE sequence to which Nrf2 binds not pointed out. Activation of Nrf2 leads (line 77) not only to expression of genes encoding antioxidant defense (line 77) but several others affecting carcinogens metabolism or even cell proliferation.
Moreover, the action of other phytochemicals is presented. Thus the title does not reflect the paper content.
Therefore, section 3 should be removed or the title changed for example as follows: Therapeutic potential of kaemferol and the other naturally occurring polyphenols might be modulated by Nrf2-ARE signaling pathway.
The classification of polyphenols should be provided. Table 1 Phytochemicals mentioned are not nutrient! In opposite they are non-nutrient plant components.
Line 156 - kaemferol is mentioned. This information should be placed in section 4.
Section 4 paragraph between the lines 162-168 is the repetition of statements from Introduction section. This section should be rather combined with that describing potential chemopreventive or therapeutic effects of kaempferol in the sections addressed to particular diseases. Concerning cancer, it should be clearly stated its role in cancer prevention and possible role in adjuvant cancer therapy.
Several quotations in the text are not precise and should be improved. For example, in paragraph between lines 234-244 the effects of Modified Xiaoyao powder are described, but its content is not mentioned.
Similarly, the link of estrogens (lines 274-282) with Nrf2 requires better explanation.
The section 6 should be removed. In review papers the Discussion section is not predicted, and in this case mostly repeats the earlier statements.
In general, the paper is poorly written (including English usage). There are many repetitions, the paper lacks of flow what certainly is the result of involvement too many authors.
Figures might be combined and their descriptions should be more detailed.
Abbreviations are not consequently used (e.g., ROS).
Author Response
Dear Reviewer
Many thanks for your valuable suggestions.
This review authorized by 7 authors from different countries aimed to describe the role of Nrf2 pathway in chemopreventive or therapeutic potential of kaempferol.
However, this manuscript is actually more concentrated on the description of Nrf2 pathway, although not in details. In this regard the mechanisms of its activation, particularly non-canonical, are not well presented or explained. In Figure 1 MAF protein is shown, but its role in Nrf2 is not provided and ARE sequence to which Nrf2 binds not pointed out. Activation of Nrf2 leads (line 77) not only to expression of genes encoding antioxidant defense (line 77) but several others affecting carcinogens metabolism or even cell proliferation.
Author response:
The mentioned suggestions were carried out as recommended. The changes made were highlighted within the manuscript.
Moreover, the action of other phytochemicals is presented. Thus, the title does not reflect the paper content. Therefore, section 3 should be removed or the title changed for example as follows: Therapeutic potential of kaemferol and the other naturally occurring polyphenols might be modulated by Nrf2-ARE signaling pathway.
Author response:
Title of the article was modified as per reviewer instructions.
The classification of polyphenols should be provided. Table 1 Phytochemicals mentioned are not nutrient! In opposite they are non-nutrient plant components.
Author response:
Followed as suggested, changes made were highlighted.
Line 156 - kaemferol is mentioned. This information should be placed in section 4.
Author response:
Changes were made as suggested by reviewer.
Section 4 paragraph between the lines 162-168 is the repetition of statements from Introduction section. This section should be rather combined with that describing potential chemopreventive or therapeutic effects of kaempferol in the sections addressed to particular diseases. Concerning cancer, it should be clearly stated its role in cancer prevention and possible role in adjuvant cancer therapy.
Author response:
Changes were as made as suggested and were highlighted within the manuscript.
Several quotations in the text are not precise and should be improved. For example, in paragraph between lines 234-244 the effects of Modified Xiaoyao powder are described, but its content is not mentioned.
Author response:
Changes were made to the article as suggested. Contents of Xiao powder were mentioned and highlighted.
Similarly, the link of estrogens (lines 274-282) with Nrf2 requires better explanation.
Author response:
It was explained and highlighted.
The section 6 should be removed. In review papers the Discussion section is not predicted, and in this case mostly repeats the earlier statements.
Author response:
The discussion section was removed
In general, the paper is poorly written (including English usage). There are many repetitions, the paper lacks of flow what certainly is the result of involvement too many authors.
Author response:
The manuscript was revised thoroughly and crosschecked again.
Figures might be combined and their descriptions should be more detailed.
Author response:
Suggestions were fulfilled as per reviewer instructions
Abbreviations are not consequently used (e.g., ROS).
Author response:
It was done as suggested
Reviewer 2 Report
The mansucript “Kaempferol and Its Therapeutic Potential Modulated by Nrf2: Current Status and Future Direction” by Hussain et al. reviews the potential active properties of kaempferol on the regulation of oxidative stress in different diseases through the activation of the Nrf2 signaling pathway.
The authors gathered information on the effects of kaempferol on Nrf2 modulation in different cell cultures and in vivo models. They also give some background on the effects of regulating Nrf2-related cascades. Nonetheless, some changes should be made before the manuscript is considered for publication:
- The authors should better explain the role of Nrf2 in cancer, diabetes, and cardiovascular disease…
- The authors need to include more information on the effects of kaempferol on other diseases such as obesity or other metabolic-associated deseases… They must include the most updated literature review of the effects of kaemferol and the regulation on Nrf2. There are still some manuscripts that could be included and would complete the review.
- The authors should revise the grammar.
- The authors should rewrite in vitro and in vivo.
- Kaempferol is a poorly absorbed flavonoid. It is also modified during digestion and colonic fermentation. Then, some of the effects ascribed to kaempferol in vivo may derive from other catabolites. The authors should discuss this aspect in their discussion.
Author Response
Dear Reviewer
Many thanks for your valuable suggestions.
The mansucript “Kaempferol and Its Therapeutic Potential Modulated by Nrf2: Current Status and Future Direction” by Hussain et al. reviews the potential active properties of kaempferol on the regulation of oxidative stress in different diseases through the activation of the Nrf2 signaling pathway.
The authors gathered information on the effects of kaempferol on Nrf2 modulation in different cell cultures and in vivo models. They also give some background on the effects of regulating Nrf2-related cascades. Nonetheless, some changes should be made before the manuscript is considered for publication:
- The authors should better explain the role of Nrf2 in cancer, diabetes, and cardiovascular disease…
Author response:
Changes were made as suggested by the reviewer and were highlighted within the manuscript.
- The authors need to include more information on the effects of kaempferol on other diseases such as obesity or other metabolic-associated deseases… They must include the most updated literature review of the effects of kaemferol and the regulation on Nrf2. There are still some manuscripts that could be included and would complete the review.
Author response:
Changes were made and addition was carried out as suggested. However, the effect of kaempferol on obesity targeting Nrf2 was added best to our knowledge from the latest data published.
- The authors should revise the grammar.
Author response:
It was revised thoroughly
- The authors should rewrite in vitro and in vivo.
Author response:
The invitro and invivo data was revised and explained.
- Kaempferol is a poorly absorbed flavonoid. It is also modified during digestion and colonic fermentation. Then, some of the effects ascribed to kaempferol in vivo may derive from other catabolites. The authors should discuss this aspect in their discussion.
Author response:
The discussion portion was removed as per instruction of another reviewer.
Reviewer 3 Report
In this review the authors highlight the therapeutic potential of kaempferol, which acts through the regulation of Nrf2.
Phytochemicals are compounds that have great therapeutic potential and this review offers valid points for reflection.
Nrf2 is also a molecule that is much studied and appreciated in the oncolological and therapeutic fields in general. I advise authors to rate the work Int J Mol Sci. 2021 Jul 26; 22 (15): 7963. doi: 10.3390 / ijms22157963.
Paragraph 2 (Nrf2 modulation) is very generic. The authors refer to whether Nrf2 activates or is activated by proteins, kinases, without mentioning at least one.
This paragraph should be richer and more subdivided (it is not clear if the authors want to talk about how Nrf2 is regulated or the role of Nrf2 in different pathologies).
The authors in the figure insert MAF, but I have not found in the text.
Author Response
Dear Reviewer
Many thanks for your valuable suggestions.
In this review the authors highlight the therapeutic potential of kaempferol, which acts through the regulation of Nrf2.
Phytochemicals are compounds that have great therapeutic potential and this review offers valid points for reflection.
Nrf2 is also a molecule that is much studied and appreciated in the oncolological and therapeutic fields in general. I advise authors to rate the work Int J Mol Sci. 2021 Jul 26; 22 (15): 7963. doi: 10.3390 / ijms22157963.
Author response:
The mentioned article was discussed and cited within the manuscript.
Paragraph 2 (Nrf2 modulation) is very generic. The authors refer to whether Nrf2 activates or is activated by proteins, kinases, without mentioning at least one.This paragraph should be richer and more subdivided (it is not clear if the authors want to talk about how Nrf2 is regulated or the role of Nrf2 in different pathologies).
Author response:
New data was added as per reviewer instructions. Role of Nrf2 was added to the concerned pathology section in the manuscript. Changes made were highlighted.
The authors in the figure insert MAF, but I have not found in the text.
Author response:
The mentioned changes were made to the manuscript in correspondence with the figure.
Round 2
Reviewer 1 Report
My previous review was concluded with the following statement:
“In general, the paper is poorly written (including English usage). There are many repetitions, the paper lacks of flow…”.
While in the revised manuscript the authors addressed my specific comments, the paper body was not changed and the above remark is still topical.
Definitely, the paper requires professional proofreading to correct English usage.
Besides, the authors should address yet some specific comments listed below:
Paragraph between the lines 59-61 along with the reference should be removed as the reference content is not cited correctly.
Section 2: Should be entitled: Nrf2 activation/inhibition – pharmacological target instead of modulation.
Figure 1 caption: Various molecular mechanisms in cancer cells lead to activation of Nrf2 instead of constitute activation. This is just one example of not correct English usage.
Paragraph between the lines 106-109
Should be changed as follows: Nrf2 is the key transcription factor regulating antioxidant and xenobiotic exposure response. When exposed to oxidative stress or cells’ potential damaging agents, Nrf2 translocate to cell nucleus and forms heterodimer with small Maf (sMaf) proteins. Nrf2/smaf heterodimer binds specifically to a cis -acting enhancer called antioxidant response element (ARE) and initiates transcription of genes encoding antioxidant and detoxification proteins. (Wenge et al. BBA,2008).
Line 138 Flavonoids and phenolic acids should be in plural similarly as Phytochemicals in Figure 2. Caption: ARE is not signaling pathway but enhancer or Nrf2 target DNA sequence.
Table 1 Phytonutrient in the first column was not change into phytochemical as was requested for the reason explained before.
Section 4
The sentence: The main sources of kaempferol include tea, kale, spinach, grapes and gingko biloba leaves should be removed since repeats the information from the Introduction.
Kaempferol flavanol (page 7) or flavonoid (page 2)? Kaempferol represents Flavonoids subclass-Flavanols. It should be clarified.
Line 211 the word “production” (ROS) is missing.
Section 5: Should be entitled: Kaemperol affects Nrf2 activation in different pathological conditions/diseases
Line 245
“In combating cancer, kaempferol regulates transcriptional pathways of Nrf2 as well as reduces redox homeostasis at cellular level. Such defined mechanisms are shown in Figure 4. “
What does it mean reduces redox homeostasis? It is rather expected that compounds such as kaempferol should restore redox homeostasis.
In Figure 4 and in the text the double face of Nrf2 activation should be addressed. For cancer treatment inhibitors of Nrf2-ARE pathways are required.
Description of this figure should be changed, similarly as Figure 5, which in current form is unclear.
Figure 6
The content of the Upper Left box as well as whole figure’s description does not make sense. In fact, this figure might be removed, because does not provide anything new.
Section 6
Many statements in this section are oversimplification. Other effects of Nrf2 activation or inhibition besides ROS regulation should be at least mentioned.
Author Response
Dear Reviewer
Many thanks for the valuable comments.
In general, the paper is poorly written (including English usage). There are many repetitions, the paper lacks of flow…”.
Carefully read and corrected
Author Response: The extensive corrections have been made by Prof. Dr. Michael Aschner in order to correct the grammatic mistakes and improve flow.
While in the revised manuscript the authors addressed my specific comments, the paper body was not changed and the above remark is still topical.
Author Response: Needful corrections and additions are done to make it attractive.
Definitely, the paper requires professional proofreading to correct English usage.
Author Response: The extensive corrections have been made by Prof. Dr. Michael Aschner in order to correct the grammatic mistakes and improve flow.
Carefully read and corrected
Author Response: Needful is done.
Besides, the authors should address yet some specific comments listed below:
Paragraph between the lines 59-61 along with the reference should be removed as the reference content is not cited correctly.
Author response: Removed as suggested
Section 2: Should be entitled: Nrf2 activation/inhibition – pharmacological target instead of modulation
Author response: Changed as per suggestions.
Figure 1 caption: Various molecular mechanisms in cancer cells lead to activation of Nrf2 instead of constitute activation. This is just one example of not correct English usage.
Author response: Corrected
Paragraph between the lines 106-109
Should be changed as follows: Nrf2 is the key transcription factor regulating antioxidant and xenobiotic exposure response. When exposed to oxidative stress or cells’ potential damaging agents, Nrf2 translocate to cell nucleus and forms heterodimer with small Maf (sMaf) proteins. Nrf2/smaf heterodimer binds specifically to a cis -acting enhancer called antioxidant response element (ARE) and initiates transcription of genes encoding antioxidant and detoxification proteins. (Wenge et al. BBA,2008).
Author response: Changed as suggested
Line 138 Flavonoids and phenolic acids should be in plural similarly as Phytochemicals in Figure 2. Caption: ARE is not signaling pathway but enhancer or Nrf2 target DNA sequence.
Author response: Changes made as suggested by the reviewer
Table 1 Phytonutrient in the first column was not changed into phytochemical as was requested for the reason explained before.
Author response: the word changed to suggested one
Section 4
The sentence: The main sources of kaempferol include tea, kale, spinach, grapes and gingko biloba leaves should be removed since repeats the information from the Introduction.
Author response: Removed
Kaempferol flavanol (page 7) or flavonoid (page 2)? Kaempferol represents Flavonoids subclass-Flavanols. It should be clarified.
Author response: Changes were made. There was topographic mistake that was removed.
Line 211 the word “production” (ROS) is missing.
Author response: the word production was added
Section 5: Should be entitled: Kaemperol affects Nrf2 activation in different pathological conditions/diseases
Author response: the title was changed as per suggestions
Line 245
“In combating cancer, kaempferol regulates transcriptional pathways of Nrf2 as well as reduces redox homeostasis at cellular level. Such defined mechanisms are shown in Figure 4. “
Author Response: Needful changes have been made.
What does it mean reduces redox homeostasis? It is rather expected that compounds such as kaempferol should restore redox homeostasis.
Author response: Corrected
In Figure 4 and in the text the double face of Nrf2 activation should be addressed. For cancer treatment inhibitors of Nrf2-ARE pathways are required.
Description of this figure should be changed, similarly as Figure 5, which in current form is unclear.
Author Response: Needful changes have been made.
Figure 6
The content of the Upper Left box as well as whole figure’s description does not make sense. In fact, this figure might be removed, because does not provide anything new.
Author response: Removed
Section 6 (Now section 5)
Many statements in this section are oversimplification. Other effects of Nrf2 activation or inhibition besides ROS regulation should be at least mentioned.
Response: Needful changes have made.
Regards
Reviewer 2 Report
As previously stated, Kaempferol is a poorly absorbed flavonoid. It is also modified during digestion and colonic fermentation. Then, some of the effects ascribed to kaempferol in vivo may derive from other catabolites. The authors should discuss this aspect at the end of their manuscript. Create a new section.
Author Response
Dear Reviewer
Many thanks for your valuable suggestions.
As previously stated, Kaempferol is a poorly absorbed flavonoid. It is also modified during digestion and colonic fermentation. Then, some of the effects ascribed to kaempferol in vivo may derive from other catabolites. The authors should discuss this aspect at the end of their manuscript. Create a new section.
Author response: A short section has been added as per the suggestions of the reviewer coupled with the scope of the article.
Regards